# Technical Note: Testing the Connection Between Hillslope Scale Runoff Fluctuations and Streamflow Hydrographs at the Outlet of Large River Basins

Ricardo Mantilla[1], Morgan Fonley[2], and Nicolás Velasquez[3]

[1]Department of Civil Engineering, University of Manitoba, Winnipeg, MB, Canada
[2]Alma College, Alma, MI, USA
[3]Iowa Flood Center, University of Iowa, Iowa City, IA, USA

*Correspondence to*: Ricardo Mantilla (ricardo.mantilla@umanitoba.ca)

**Abstract.** A series of numerical experiments were conducted to test the connection between streamflow hydrographs at the outlet of large watersheds and the time-series of hillslope-scale runoff yield. We used a distributed hydrological routing model that discretizes a large watershed (~17,000 km$^2$) into small hillslope units (~0.1 km$^2$) and applied distinct surface runoff time-series to each unit that deliver the same volume of water into the river network. The numerical simulations show that distinct runoff delivery time-series at the hillslope scale result in indistinguishable streamflow hydrographs at large scales. This limitation is imposed by space-time averaging of input flows into the river network that are draining the landscape. The results of the simulations presented in this paper show that under very general conditions of streamflow routing (i.e., nonlinear variable velocities in space and time), the streamflow hydrographs at the outlet of basins with Horton-Strahler (H-S) order five or above (larger than 100 km$^2$ in our set up) contain very little information about the temporal variability of runoff production at the hillslope scale and therefore the processes from which they originate. In addition, our results indicate that the *rate of convergence* to a common hydrograph shape at larger scales (above H-S order 5) is directly proportional to how different the input signals are to each other at the hillslope scale. We conclude that the ability of a hydrological model to replicate outlet hydrographs does not imply that a correct and meaningful description of small-scale rainfall-runoff processes has been provided. Furthermore, our results provide context for other studies that demonstrate how the physics of runoff generation cannot be inferred from output signals in commonly used hydrological models.

## 1 Introduction

The question of how will river flows change in the presence of climatic and anthropogenic changes dominates the literature in water resources journals and research conducted around the world (Arnell & Lloyd-Hughes, 2014; Blöschl et al., 2019; Gudmundsson et al., 2021; Hirabayashi et al., 2013; Whitehead et al., 2018). Slowly evolving regional climatic changes and rapid anthropogenic modifications to the landscape will impact how watersheds deliver water to communities along the river network, and recognition of this fact has fostered the development of methods and techniques to address this urgent problem (Kang et al., 2016; Kourtis & Tsihrintzis, 2021). Physics-based hydrological modeling has emerged as the preferred alternative for predicting the future of the hydrological cycle under the projected changes (Barnett et al., 2008). An example of rapid anthropogenic change in the US Midwest is the use of drainage tiling, which is effective for increasing corn yields by promoting the rapid movement of subsurface flows into rivers (Fonley et al., 2021; Schilling et al., 2019; Schilling & Helmers, 2008; Velásquez et al., 2021). This raises the question of how these landscape modifications will impact the cycles of flooding and droughts in future climatic scenarios (Akter et al., 2018; Júnior et al., 2015; Peng et al., 2019). In the typical approach to this question, a land-surface model is

conceptualized, calibrated, validated, and then forced with past and future climatic inputs (rainfall, radiation, relative humidity) derived from global circulation models (Barnett et al., 2008; Condon et al., 2015; Hidalgo et al., 2009; Sadeghi Loyeh & Massah Bavani, 2021; Taye et al., 2015; Quintero et al. 2018). The underlying premise is that the hillslope-scale process equations correctly

describe the water movement and partitioning into runoff and infiltration. This premise allows changing parameter values to reflect future land-surface conditions (e.g., vegetation types, deforestation, land cover, land use) along with future conditions of meteorological forcing. Nevertheless, results from Huang et al. (2017) at 12 large-scale watersheds using nine models indicate significant limitations during extreme events. Some limitations can be improved by calibration at the cost of losing representation of processes such as evapotranspiration (Ahmed et al., 2023).

The use of land-surface models relies on their validation after parameter calibration (Beven, 1989). Historically, parameters in land-surface models are adjusted (calibrated) to match streamflow hydrographs at gauged sites in the outlet of large watersheds (> 100 km$^2$), and a portion of the data is left to validate the hydrological model using unobserved events (Bérubé et al., 2022; Gupta et al., 2006; Refsgaard, 1997; Shen et al., 2022). Once this test has been passed, a land-surface model is deemed appropriate to explore future scenarios (Sadeghi Loyeh & Massah Bavani, 2021; Schilling et al., 2008; Whitehead et al., 2018). Several authors

have highlighted the issue that stronger and more appropriate methods and techniques for validating/invalidating hydrologic models are needed (e.g., Beven and Lane, 2019), but the ability to simulate streamflow hydrographs continues to dominate the literature as the standard for model validation. Other data-based techniques for streamflow prediction are not appropriate to address these questions because their parameters cannot be directly linked to physical processes and because there is a recognition that non-stationarities render historical information unreliable to predict the future behavior of hydrological systems (Bayazit, 2015;

Cancelliere, 2017; Milly et al., 2008; Salas et al., 2018). Furthermore, a recent simulation based study by Remmers et al., (2020) concluded that "it is challenging and in most cases impossible to infer model structure from model output for the part of model space, bucket-based hydrological models, that [were] sampled". This again raises the historic question: what are the mechanisms that blur such differences? This issue of identifiability of hydrologic models has been raised as far back as research in the early 60's (Dawdy and O'Donnell, 1965). In this sense, different authors have arrived at similar conclusions. According to De Boer-

Euser et al. (2017), it is challenging to link hydrograph differences and model structure components. Moreover, in an intercomparison experiment of three models, Vetter et al. (2015) observed similar performance among them. Tijerina et al., (2021) described similar spatial performance of two hydrological models over the conterminous US and the Mai et al. (2022) comparison of 13 models obtained different rankings in the best-performing models when evaluating discharge versus distributed variables such as snow-water equivalent (SWE). Additionaly, in the recent evaluation of Hydrology Laboratory-Research Distributed

Hydrologic Model (HL-RDHM) by Madsent et al. (2020) it was shown that the performance of streamflow predictions was scale dependent, decreasing for small basins compared to the performance in larger basins, a result that is consistent with other multiscale evaluations of predictive performance of distributed models (Seo et al 2021, Seo et al. 2023).

There is a long history in the hydrologic literature showing that dispersion and aggregation of flows in the river network dominates the shape of the hydrograph at larger scales (Surkan, A.J., 1968; Kirkby, 1976; Beven and Wood, E. F., 1993; Snell and Sivapalan,

1994; Mantilla et al., 2006; Zarlenga et al. 2022), however, there isn't a systematic evaluation of how quickly the river network effects dominate over hillslope scale variability under generic conditions of routing and heterogeneity among hillslopes. In the present work we explore how different hillslope scale (~ 0.1 km$^2$) surface runoff time-series correspond to hydrographs at the outlet of larger scales (> 100 km$^2$) basins. We used a distributed hydrological model that discretizes the landscape into small hillslope control volumes interconnected by the full extent of the river network of the Cedar River basin (~17,000 km$^2$). We created

a set of forcing signals that are significantly different from each other but share the same volume of water injected into the hillslope surface. Each of the input signals represents the output of distinct land-surface process descriptions. Our results show that input

signals that are very different from each other at the hillslope scale produce streamflow hydrographs at large scales that are indistinguishable from each other. This result confirms that our ability to reproduce hydrographs at the outlet of a large basin is not a reliable indicator that we have correctly described small-scale processes controlling runoff production that include the description of vegetation, soil types, land use practices, snowmelt rates, etc. Furthermore, we provide a quantative prediction for how quickly the information of small scale processes dissipate in the river network, showing why it is difficult, if not imposible, to establish a causal link between runoff generation mechanisms and output hydrographs as discussed by Remmers et al., (2020). Finally, our work provides quantitative measures that prevent linking the properties of outlet hydrographs to plot scale process that can prevent accepting model descriptions that can be right for the wrong reasons (Kirchner, J.W., 2006).

Our paper is organized as follows: In section 2, we present the main methodological steps of our work, including the description of the routing model, the river network used, the input signals representing surface runoff, and the metrics for comparison of similarity between hydrographs.  In section 3, we present the main results of the simulations for three levels of complexity in the description of flow routing. Section 4 discusses the results and how they furnish the basis for the conclusions presented in Section 5.

## 2 Methodology and Data

### 2.1 The Hydrological Model & The River Network

The distributed hydrological model used in this study can be succinctly described as a nonlinear water transport equation through a directed river, which is the routing component of the operational flood forecasting model HLM (Mantilla et al., 2022). The model is formulated in the context of a mass conservation equation developed by Gupta and Waymire (1998), and uses the water velocity parameterization given by Mantilla (2007) where flow velocity for any link $j$ in the river network at time $t$ is given by $v(j,t) = v_o q_{j,n}^{\lambda_1} A_j^{\lambda_2}$, where $A_j$ in $[km^2]$ is the upstream basin area from link $j$, and $q_{j,n}$ in $[m^3/s]$ is the flow from the channel link $j$ toward the downstream channel link $n$, which is further related to water storage in the link $s_c^{(j)}$ by the relation $q_{j,n} = \frac{v(j,t)}{l_j} s_c^{(j)}(t)$.

The equation is non-homogeneous, therefore, the velocity $v_o$ given in $[m/s]$ is a reference quantity that corresponds to the flow velocity for a channel link that drains a $1\ km^2$ catchment when $1\ m^3/s$ flows through. The non linear scaling relationship of velocities given by the parameters $\lambda_1$ and $\lambda_2$ are derived from assumptions about the variation of local and downstream variation in hydraulic geometry (cross sectional area and friction) with respect to basin area that are well established in the literature (Singh 2022). The parameterization of velocity replaces the momentum conservation equation (see Mantilla (2007) for details) and is equivalent to the kinematic wave simplification of the Saint-Venant equations integrated over the channel length. The parameterization is given by,

$$\frac{dq_{j,n}(t)}{dt} = \frac{v_o q_{j,n}^{\lambda_1}(t) A_j^{\lambda_2}}{l_j(1-\lambda_1)}\left(q_{p,c}(j,t) - q_{j,n}(t) + \sum_{f \to j} q_{f,j}(t)\right) \tag{1}$$

The index $f$ in the equation refers to the set of upstream links draining into link $j$. The parameterization is equivalent to assuming that the channel slope is equal to the slope of the water surface (normal flow) and that the flow is purely kinematic, obeying a power law relation between flow depth and velocity (such as Manning, Chezy, or the Darcy-Weisbach equations). As pointed out by Beven (1979) there is distinction between flow velocity and celerity in the system. In our equation celerity $c$ is related to flow velocity by the relationship $c(j,t) = \frac{v(j,t)}{(1-\lambda_1)}$ making celerity spatially and temporally variable. In particular, for the three scenarios of flow velocity that are investigated in this paper (see Velasquez and Mantilla, 2020 for definitions), when velocity is assumed to be constant in space and time (i.e. $\lambda_1 = 0$ and $\lambda_2 = 0$) celerity is equal to velocity; in the case of the global self-similar scenario

where $\lambda_1 = 0.3$ celerity $c(j,t) = 1.43 \, v(j,t)$; and for the local self-similar scenario where $\lambda_1$ takes random values beteen 0.1 and 0.5, celerity takes values between $1.1 \, v(j,t)$ and $2.0 \, v(j,t)$. Finally, the function $q_{p,c}(j,t)$ represents the flow from the hillslope $j$ into the channel link $j$. For this paper, the runoff function is,

$$q_{p,c}(j,t) = q_{p,c}(j,0) + a_h^{\langle j \rangle} \int_0^t p(t-\tau) e^{-k_p \tau} d\tau \tag{2}$$

where the parameter $k_p$ is the inverse residence time in the hillslope surface when water flows at constant velocity $v_h$, i.e. $q_{p,c}(j,t) = a_h^{\langle j \rangle} k_p s_p^{\langle j \rangle} = a_h^{\langle j \rangle} \left( \frac{v_h}{l_h^{\langle j \rangle}} \right) s_p^{\langle j \rangle}$ over the hillslope length $l_h^{\langle j \rangle}$, and $p(t)$ represents the flow of water into the hillslope surface in units of [L/T] (e.g., mm/hr). In hydrological models, the function $p(t)$ represents "effective precipitation" or is the result of other physical processes represented in the land-surface model such as snowmelt, that are controlled by the the accumulation of water in the hillslope surface, including interception, evaporation, infiltration, and exfiltration.

The river network chosen for this study was the Cedar River upstream of Cedar Rapids, Iowa (Figure 1). We chose this network to maintain the realism of the connectivity structure. However, simulations not presented in this paper indicate that our results can be generalized to any random-self similar network. At the outlet, the network reaches a Horton order of 8, with a width function that shows the largest accumulation of channels at a common distance in the upper region of the watershed (Figure 1a). We also show width functions for select smaller Horton-order locations in the river network to illustrate the natural variability of river network connectivity across scales.

**2.2 Development of Hillslope-scale Runoff Fluctuations**

In general, any rainfall pattern which is integrable over an interval of length can be written in terms of a Fourier series expansion as

$$p(t) = \frac{a_0}{2} + \sum_{n=1}^N a_n \cos\left( \frac{2\pi n t}{T_r} - \varphi_n \right), \tag{3}$$

where phase shifts allow us to represent sines (typically present in the Fourier series expansion) in terms of cosines of the same period. In this format, the coefficients can be chosen to sufficiently represent any function, including those that are not smooth. We applied inverted and shifted cosines representing one, two, three, and four cosine waves distributed over the course of one or three days. The cosines were shifted to affirm they were nonnegative and inverted—beginning and ending with zero precipitation intensity—to impose smoothness on the rainfall signal. We included both right-skewed and left-skewed sawtooth (linear) patterns and signals depicting uniform precipitation over 1, 2, 3, and 6 days.

Input to the model was applied as flux into a hillslope surface represented by a linear reservoir that feeds into the adjacent river link. Each input waas designed to deliver 48 mm of water to the hillslope, and each signal began at time zero. Flow in the river network was primed by simulating a one-day rainfall event and allowing the system to drain for ten days before applying the rainfall signal so that each link of the river network had an initial flow that was commensurate with the watershed it drained.

**2.3 Developing a Reference Hydrograph for Each Location in the River Network**

Comparing hydrographs across scales can be difficult due to the variability involved. To address this difficulty, we created dimensionless hydrographs for all locations in the river network with the following equations:

$$q^* = \frac{q}{q_{ref}} \qquad \text{and} \qquad t^* = \frac{t}{t_{ref}} \tag{4}$$

where $q_{ref}$ is a reference maximum streamflow and $t_{ref}$ is a reference hydrograph time to peak. For reference we chose the instantaneous unit hydrgraph that resulted from applying precipitation via a delta-Dirac function, such that the volume of water was directly applied as an initial condition onto the hillslope surface, acting as an instantaneous event. From that hydrograph, we

identified the peak flow and the time to peak flow for every link in the river network. We defined an event's 'end' as the moment when the flow was smaller than 1% of the peak flow.

## 2.4 Analysis of Convergence of Streamflow Signals at Larger Scales

To analyze the convergence of the hydrographs for streams of order $\omega$, we compared moments about the time-axis ($t^*$) and about the streamflow-axis ($q^*$). We computed the first moment in $t^*$ and in $q^*$ as

$$M_1^{t^*}(\omega) = \frac{\int_0^{T^*} t^* \cdot q^*(t^*) dt^*}{\int_0^{T^*} q^*(t^*) dt^*} \tag{5}$$

$$M_1^{q^*}(\omega) = \frac{\int_0^{T^*} q^*(t^*) dt^*}{\int_0^{T^*} t^* dt^*}, \tag{6}$$

and the second-centered moment with the following equations,

$$M_2^{t^*}(\omega) = \frac{\int_0^{T^*} (t^* - M_1^{t^*})^2 \cdot q^*(t^*) dt^*}{\int_0^{T^*} q^*(t^*) dt^*} \tag{7}$$

$$M_2^{q^*}(\omega) = \frac{\int_0^{T^*} (q^* - M_1^{q^*})^2 dt^*}{\int_0^{T^*} t^* dt^*} \tag{8}$$

Using these definitions of moments, a ratio of the moments between two different hydrographs can be defined as

$$R_{i(A|B)}^Z(\omega_j) = \frac{max\left(M_i^Z(\omega_j)_A, M_i^Z(\omega_j)_B\right)}{min\left(M_i^Z(\omega_j)_A, M_i^Z(\omega_j)_B\right)} \tag{9}$$

where $Z$ represents the variable of interest (in this case $q^*$ and $t^*$). If the moments of two different functions, A and B, approach each other, then equation (9) tends to one in the limit as,

$$\lim_{\omega \to \infty} R_{i(A|B)}^Z(\omega) = 1 \tag{10}$$

The min and max functions in equation (8) guarantee that the limit will be reached from above in all cases. This property allowed us to define a sigmoid function to characterize the rate of convergence to the limit of $\omega$ as,

$$\hat{R}_{i(B)}^Z(\omega) = S_f \cdot \left(1 - \frac{1}{1 + e^{(\omega_i - \omega \cdot \alpha)}}\right) \tag{11}$$

where $S_f$ is a scale factor, $\omega_i$ is the starting Horton order, and $\alpha$ is a parameter controlling the shape of the curve. Larger values of $\alpha$ indicate rapid convergence of the curve to 1 as the Horton-order of the basin increases, and conversely, small values of $\alpha$ indicate slower convergence to 1.

## 3 Results

For each of the proposed input signals, we integrated the HLM routing equations using different assumptions about the distribution of velocity in space and time, including linear routing with constant velocity in space and time, nonlinear global self-similar routing, and locally self-similar nonlinear routing (Velasquez & Mantilla, 2020). For each setup, we obtained hydrographs at all the watershed links (43,000 in total). At the smaller scales ($\omega = 2, 3,$ or $4$), each signal produced a distinct hydrograph. However, the hydrographs became indistinguishable with increasing Horton order.

## 3.1 Comparing hydrographs at multiple scales

In Figure 2, we show the resulting hydrographs at multiple scales for input signals with a duration of 3 days. The top-left panel in Figure 2 shows the different input signals used in this experiment. We chose locations in the river network corresponding to

180 complete order streams of orders 2 to 8 to illustrate differences in hydrograph shapes at different scales. The first set of graphs shows the hydrographs calculated under the assumption of constant flow velocity in the channels (i.e. $\lambda_1 = 0$ and $\lambda_2 = 0$) for locations with increasing Horton order. We repeated the calculation using more complex nonlinear routing assumptions. In the global self-similar case, we assumed $v_0$, $\lambda_1$ and $\lambda_2$ to be the same in the entire network (yellow shade panels in Figure 2), while in the local self-similar case, the same parameters are variable throughout the catchment (green shaded panels in Figure 2).

## 3.2 Convergence

In Figure 2, we illustrate how all the 3-day duration signals tend to converge. Moreover, we can define the convergence of two or more signals relative to any signal by comparing their moments. Figure 3 shows the average value of $R_{A|B}$ for the first (circles) and the second (triangles) moments when we compare moment ratios between the LTri3Day signal with the 3DayUniform one. In both dimensions (time and magnitude, Figures 3a and b, respectively), the value of $R_{A|B}$ tends towards one with the increase of the
190 Horton order. In the supplemental material, we illustrate how the convergence of hydrographs occurs for all links in the network (Figure S1). The convergence of hydrographs also holds for the more complex cases of nonlinear routing under the assumption of global and local self-similarity.

## 3.3 General convergence to the Instantaneous Hydrograph (Dirac simulation)

To obtain a more general idea of the convergence, we computed $R_{A|B}$ for all the input signals using the Dirac function as a reference.
$R_{A|B}$ at each Horton order $\omega$ was the median value obtained for all the links of order $\omega$. The dots in Figure 4 correspond to the computed median value of $R_{A|B}$. According to our results, each signal has a different $R_{A|B}$ for $\omega = 2$, and in all the cases, tends to converge toward 1 with increasing values of $\omega$. Moreover, the convergence rate increases as a function of the initial differences. Signals with higher $R_{A|B}$ values at $\omega = 2$ tend to converge at a faster rate than others.

We describe the convergence rate of the median values in Figure 4 using equation (7). In the equation, we fixed the scale factor $S_f$
to 1.3, and we took $\omega_i = 2$ as the reference Horton order. Then, using the nonlinear least squares method, we adjusted the parameter $\alpha$ for a set of generated hydrographs. With this approach, we obtained a good representation of the signal's convergence starting at $\omega = 2$ (continuous lines in Figure 4).

According to Figure 4, the parameter $\alpha$ is an index of the convergence rate for any signal. Larger values of $\alpha$ correspond to faster convergence rates. Moreover, we found that $\alpha$ depends on the order of the moment, the dimension being considered ($t^*$ or $q^*$), and
205 the streamflow routing approach. In Figure 5, we present $\alpha$ vs $R_{A|B}(\omega = 2)$ for each moment order, dimension, and assumptions on the variability of velocities in space and time. In all cases, $\alpha$ was proportional to $R_{A|B}(\omega = 2)$. This means that the more different the hydrographs are at smaller scales, the faster the convergence rate will be to the Dirac shape at larger scales. The supplemental material presents convergence for the nonlinear global and local self-similarity cases (Figures S2 and S3).

We found that the lower $\alpha$ values in the ratio of the second moment in time (Figure 5b) correspond to lower values of $R_{A|B}$ (Figure
4b). Additionally, higher $\alpha$ values occur for the ratios of the second moment in $q$ (Figure 5d), which coincide with large $R_{A|B}$ values (Figure 4b). Also, we found that the convergence rates of the linear, nonlinear, and self-similar HLM setups are more alike in $t^*$ than in $q^*$.

## 4 Discussion & Conclusions

Hydrologists strive to "be right for the right reasons" when modeling the hydrologic cycle, however, the datasets available to validate hydrological models are sparse and often comprise only streamflow observations at the outlets of large catchments. Typically, hydrologic modelers calibrate and validate their models using available streamflow observations. Our study sheds some new light on the limits of this strategy and provides and explanation for the difficulty in establishing a causal link between small scale runoff generation processes and hydrograph shapes at the outlets of large river basins.

The moment ratios expressed in Equation 8 show how hydrographs generated by different hillslope scale runoff signals differ from each other across Horton order scales. Our study demonstrates that differences in the hydrographs at the hillslope scale are smoothed out for larger scales. The processes controlling the convergence rate are the spatial aggregation imposed by the self-similar river network draining the landscape and the attenuation that is controlled by the flow routing equations. Simulations not shown in this paper indicate that our results hold true for any self-similar network structure. However, a systematic analysis of how different network configurations determine the rate of convergence, and if self-similarity is a necessary condition, remain to be done. Two interesting findings of our study are that *i*) the rate of convergence of hydrographs to a common shape at larger scales is proportional to how different the input signals are at the hillslope scale, and *ii*) that the convergence is independent of the assumptions on the spatial and temporal variability of in-channel velocities imposed in the routing scheme.

Typically, validation of small-scale land-surface representations relies on the ability to reproduce streamflow hydrographs at the outlet of large catchments where historical records are available. An intermediate step in developing a site-specific hydrological model is calibrating its parameters that control the rainfall-runoff transformation (e.g., infiltration rates, hydraulic conductivity, interception rates) and runoff routing through the river network (e.g., channel and floodplain friction coefficients). Our results suggest that two very different descriptions of small-scale processes (e.g., variable saturated area vs. variable infiltration rates) can lead to equivalent hydrographs at larger scales. The fact that two competing hypotheses lead to the same result hampers the possibility of determining how changes in the small-scale process will affect streamflow hydrographs in larger catchments. Our analysis show that the river network connectivity leads to an averaging of the runoff produced in different locations and at different times indicating that if the right volume of runoff is applied at a given Horton-scale, the hydrographs for the network five orders or above would be indistigushible.

The generic routing schemes tested in this study give us confidence that our results can be generalized to any hydrological distributed model of any basin that explicitly includes a river network with Horton-Strahler order five or larger. This extends the problem of model "equifinality" (Beven, 2006) to a larger problem of "equivalent models" where distinct descriptions of hillslope scale descriptions lead to the same resulting outlet hydrograph.

## 5 Data and Software

Data and software for this research can be found at: https://doi.org/10.5281/zenodo.7083172

## Acknowledgments

This work was completed with support from the Iowa Flood Center, MID-American Transportation Center (Grant number: 69A3551747107), the Iowa Highway Research Board and Iowa Department of Transportation (Contract number: TR-699). RM also acknowledges current funding from NSERC (RGPIN-2023-04724) and Manitoba Hydro.

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

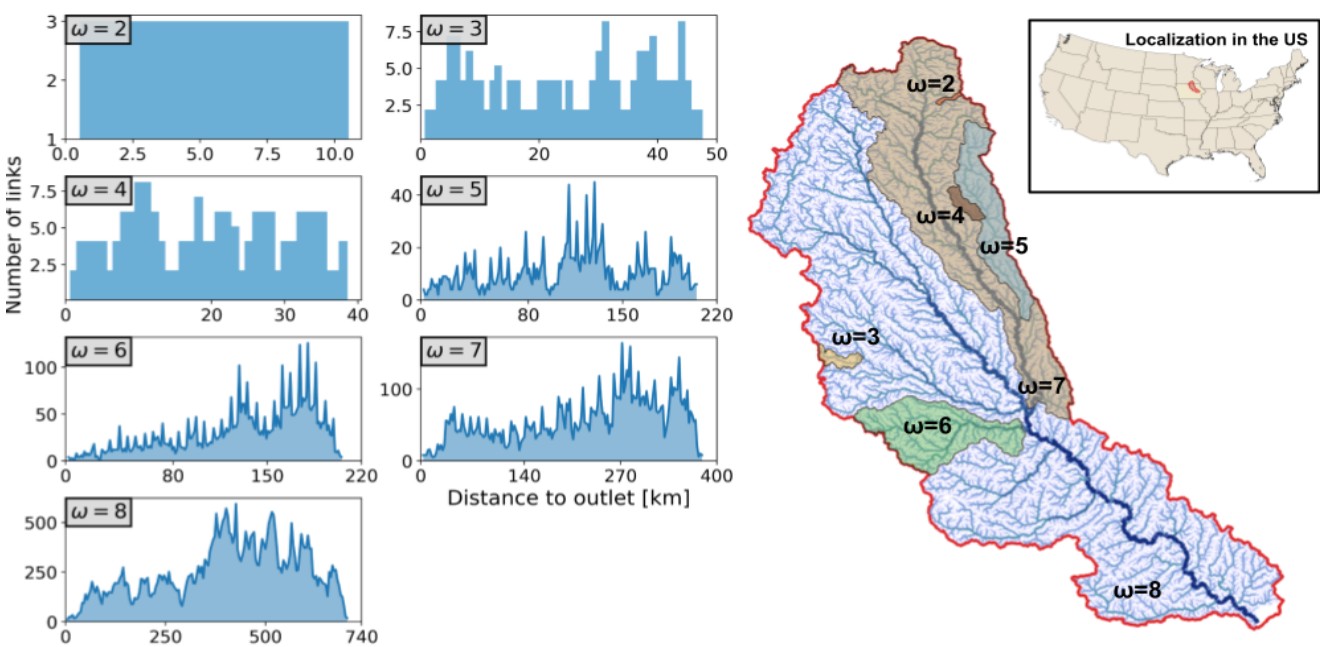

**Figure 1. Cedar River watershed network structure (right) with the width function at the outlet ($\omega = 8$) and at nested sub-watersheds with orders $\omega = 2$ to 7.**

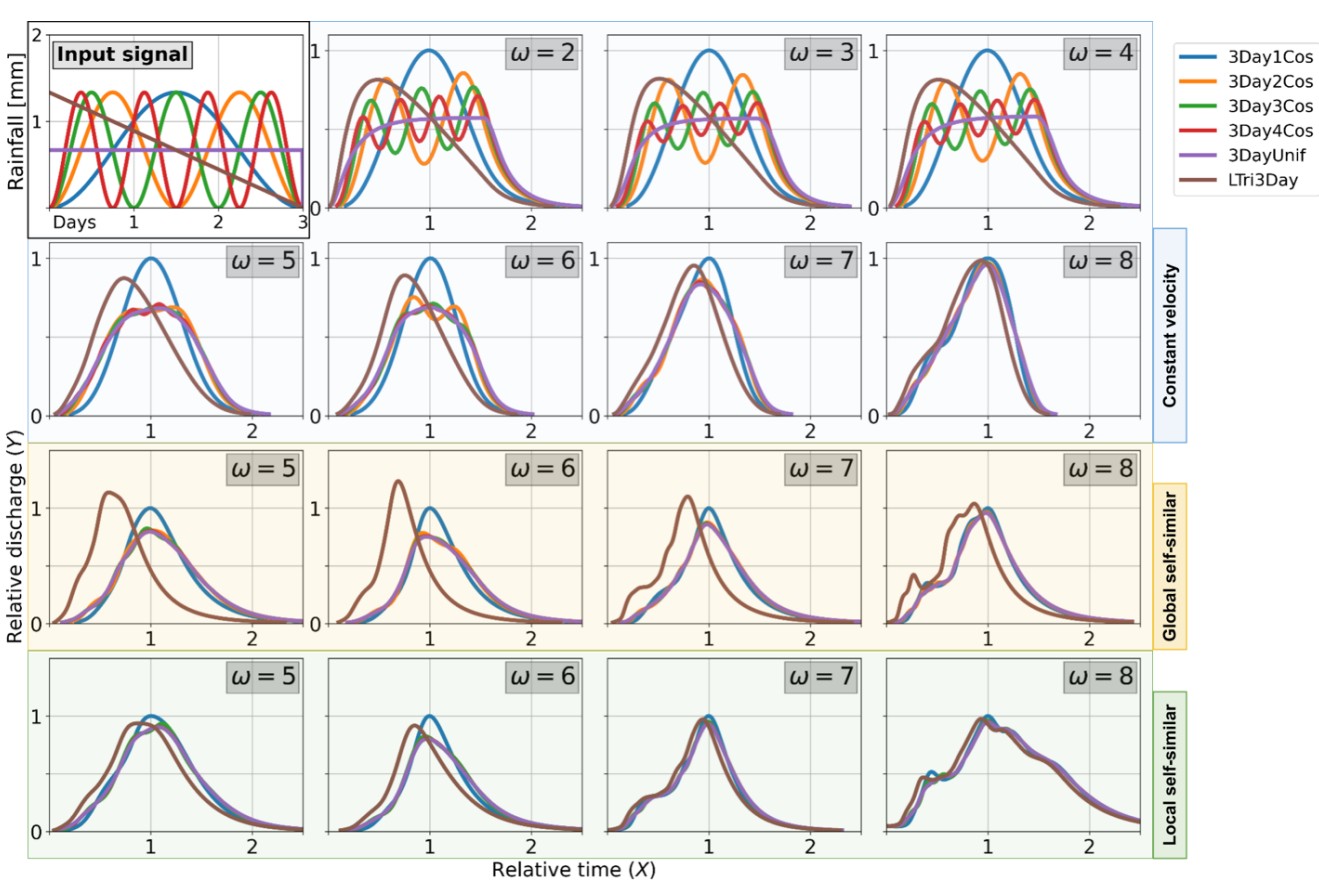

**Figure 2. Three-day duration input signals and resulting hydrographs at subwatersheds of orders between 2 and 8. The magnitude and time duration of the hydrographs is standardized relative to the 3Day1Cos (blue) hydrograph.**

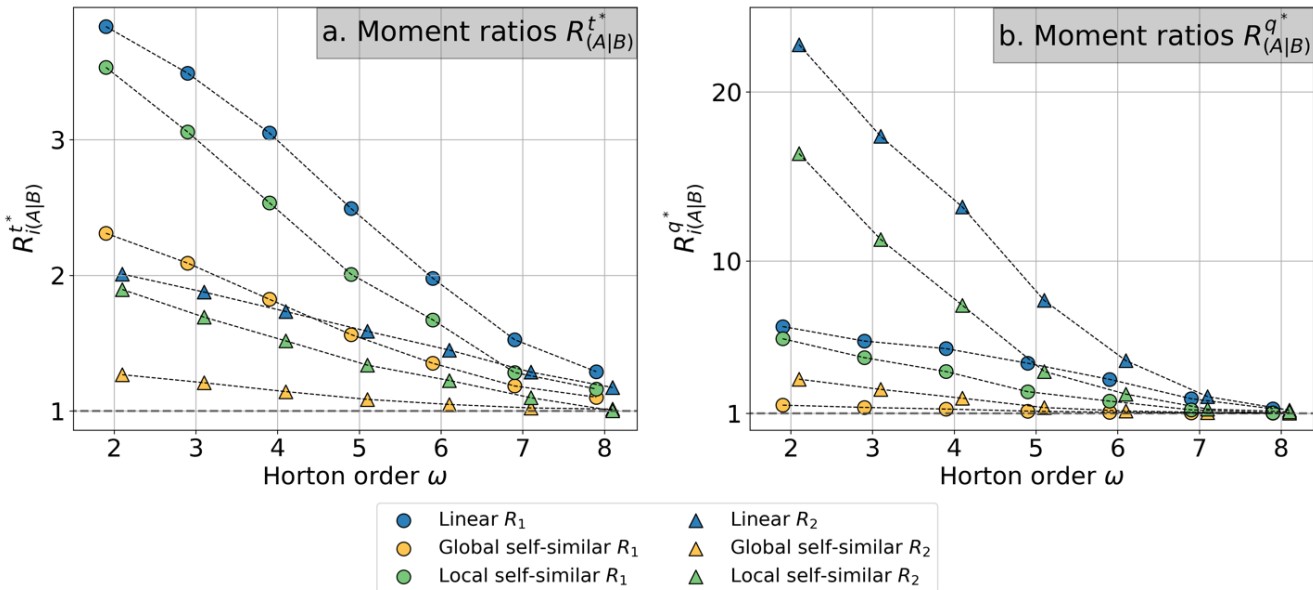

Figure 3. Changes of the first (circles) and second (triangles) moment ratios of the LTri3Day signal with respect to the 3DayUniform signal for different routing schemes. In panel a) moment ratios about the relative time $R_{i(B)}^{t^*}$, and in panel b) moment ratios about the relative streamflow $R_{i(B)}^{q^*}$.

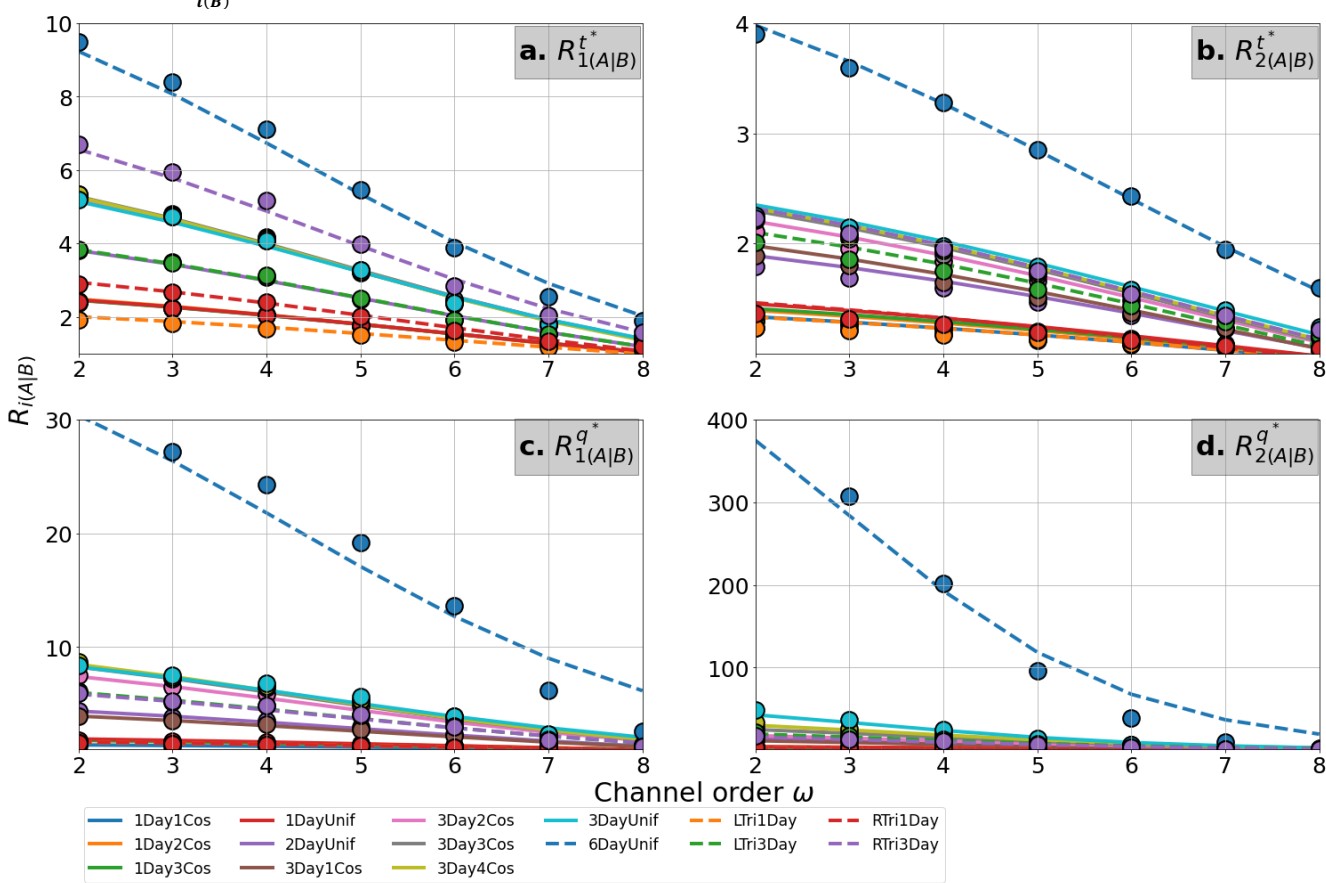

Figure 4. Convergence for all the signals relative to the Dirac assuming constant velocity routing. The dots correspond to the median value of the moments ratio $R$, and the lines correspond to equation (7) results.

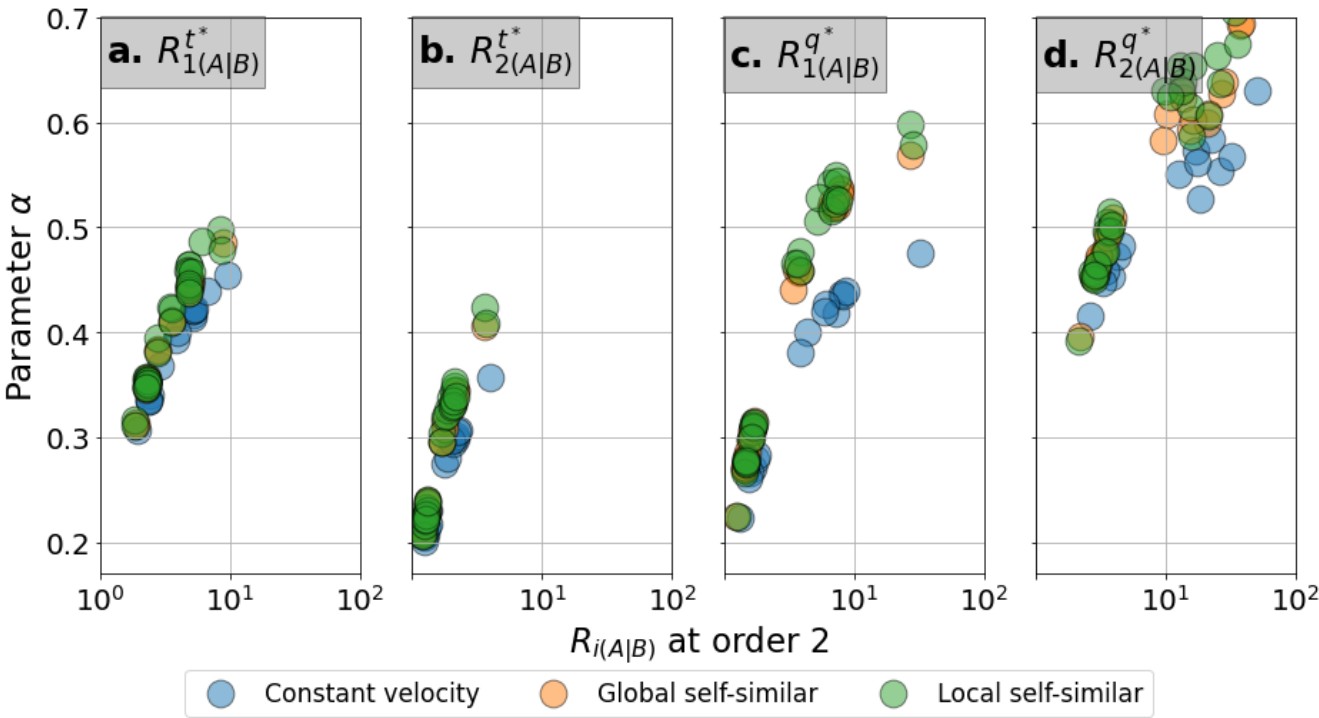

**Figure 5. Values of the estimated parameter α as a function of moment ratios for hydographs at order 2 streams for different scenarios of routing.**