# Peer review of "Technical Note: Testing the Connection Between Hillslope Scale Runoff Fluctuations and Streamflow Hydrographs at the Outlet of Large River Basins"

_Hydrology and Earth System Sciences, 2023_

## Author Comment (AC1)

According to Beven 1979

$$\frac{\partial Q}{\partial t} + C \frac{\partial Q}{\partial x} = C \, q_{\ell f(x)}$$

where $C$ is celerity (  of propagation of disturbances)

Our equation derived from continuity
and $V = V_0 \left(\frac{q}{q_{ref}}\right)^{\lambda_1} \left(\frac{A}{A_{ref}}\right)^{\lambda_2}$ where $A$ is upstream basin area

$$\frac{dQ}{dt} = \frac{V_0 \left(\frac{q}{q_{ref}}\right)^{\lambda_1} \left(\frac{A}{A_{ref}}\right)^{\lambda_2}}{(1-\lambda_1) \, \ell} \left(q_{lat} - Q + \sum Q_{up}\right)$$

Note that our equation integrates over the reach length (~500 m), but can be rewritten as

$$\frac{dQ}{dt} = \frac{V_0 \left(\frac{q}{q_{ref}}\right)^{\lambda_1} \left(\frac{A}{A_{ref}}\right)^{\lambda_2}}{(1-\lambda_1)} \left(\frac{[q_{lat} + \sum Q_{up}] - Q}{\ell}\right)$$

the term in the parenthesis can be interpreted
as $\frac{\partial Q}{\partial x}$
 & in our study $\lambda_1 = 0.3$
therefore $C = 1.43 \, V$

$$\Rightarrow C = \frac{V}{1-\lambda_1}$$

---

## Author Response (AR1)

**Response to Reviewer Comments and Recommendations for:**

**Technical Note: Testing the Connection Between Hillslope Scale Runoff Fluctuations and Streamflow Hydrographs at the Outlet of Large River Basins**

Ricardo Mantilla[1], Morgan Fonley[2], and Nicolás Velasquez[3]

[1]Department of Civil Engineering, University of Manitoba, Winnipeg, MB, Canada
[2]Alma College, Alma, MI, USA
[3]Iowa Flood Center, University of Iowa, Iowa City, IA, USA

*Correspondence to*: Ricardo Mantilla (ricardo.mantilla@umanitoba.ca)

**MS No.: hess-2023-187**

**Dear Editorial Team,**

In the text below we quote the reviewer's comments verbatim (i.e., including typographical and grammatical mistakes) and we follow them with our responses in blue.

**We have made point by point responses to your comments. We have copied your comments (in italics) followed by our response**

**Comments by Reviewer # 1 (Dr. Keith Beven)**

This is a very nice study of the effects of network dissipation on hillslope hydrographs with catchment scale but I do feel it needs some modification to reflect earlier contributions that have come to the same conclusions but which are not cited.

We start by thanking Dr. Beven for his insights regarding our work and constructive guidance on how to better contextualize our results regarding previous work on the subject. We have incomprated the body of work that has reach the same or similar qualitative conclusions regarding the dissipation of signals in river networks. We believe that our work adds specific quantitative measures regarding the rate of dissipation to support those conclusions. We recognize those contributions and make connections to our work in our revised manuscript. We expect that these changes clarify that our intention was not to claim that we "invented the wheel" but instead, that we are providing "an estimate for its diameter."

L48. But see arguments for model invalidation in Beven and Lane HP 2022

This is indeed an important point and reference. We have incorporated this reference and how it relates to the problem that we used for contextualization in Lines 50 to 53 of our revised manuscript.

L54/55. OK that is the conclusion of a specific study but why do you think this type of criticism started in 2020? Discussions about identifiability started at least as far back as Dawdy and O'Donnell in 1965, and the issues of uncertainty and equifinality on inference from models in the 1990s.

We have tried to correct this oversight in our initial manuscript. We have cited this work and its relevance in Line 59 of the new manuscript.

L60. Papers describing different optima for different objective criteria also date back to at least the 1980s, together with the discussion of Pareto optimisation for multiple criteria in the Sorooshian group at UA.

We have acknowledged this point but stopped short of including the large body of work as the focus of our paper and the results within are not related to how error is measured.

L73/74. "This result implies that our ability to reproduce hydrographs at the outlet of a large basin is not a reliable indicator that we have correctly described small-scale processes controlling runoff production". But you do not need a model to understand that the catchment is a dissipative integrative system that will necessarily make disaggregation to smaller scales highly uncertain. This was understood back in the 1970s e.g. in Kirkby, 1976, Tests of the random network model, and its application to basin hydrology, Earth Surface Processes 1 (3), 197-212 who built on the earlier work of Surkan, A. J. (1968). 'Synthetic hydrographs: effects of network geometry', *Water Resources Research*, 5, pp. 112–128. There was further work on this

in the 1990s, e.g. Beven, K.J. and E.F. Wood (1993), Flow routing and the hydrological response of channel networks, in K.J. Beven and M.J.Kirkby (eds), *Channel Network Hydrology*,Wiley,99-128. None of this earlier work is referenced in this paper.

This is a very important remark and we have cited this work and its relevance to the points we make in our manuscript. We cited this historically relevant work on dispersion in river networks and contextualize our results in the larger context of literature in Lines 65 to 73.

L85/86. Is it not celerity rather than velocity that you need for the routing (and you can have a constant celerity with a highly nonlinear velocity for certain functions – see Beven, 1979, 'On the generalised kinematic routing method'. *Water Resources Research*, 15(5), 1238-1242. Also for routing on hillslopes – see the misconception of time of concentration in Beven, K. J. 2020, A history of the concept of time of concentration, *Hydrology and Earth System Sciences*, 24: 2655–2670, doi:10.5194/hess-24-2655-2020. The manuscript should clearly differentiate between the relationship between velocities and celerities in their routing.

This is an important point. We have clarified how our formulation for flow velocity in the channels relates to celerity in our equations in Lines 130 to 149. In our formulation $v = v_0 \left(\frac{q}{q_{ref}}\right)^{\lambda_1} \left(\frac{A}{A_{ref}}\right)^{\lambda_2}$ where $q$ is discharge and $A$ is the upstream area of the link. Celerity can be derived to give c=$\frac{v}{(1-\lambda_1)}$. The value of $\lambda_1$ chosen in our numerical study is 0.3 which implies that c = 1.43 v. We added an explicit description of the treatment of celerity for all the routing cases investigated in our paper. Also, we have attached a short write up connecting the velocity and celerity in our equations (see image at the end of the response to reviewer 1).

L161. See Kirkby 1976 or Beven and Wood, 1993 again for same conclusions.

These references have been added in the appropriate context.

L199. See Beven and Lane HP 2022 for discussions of this point given uncertainties in data as well as the disaggregation problem. The equifinality concept has long suggested that you cannot differentiate between "successful/acceptable/behavioural" models but you have not referenced that either. So it does appear as if you are somewhat reinventing the wheel in both methods and conclusions.

We have tried to remove any implication that we discovered the core qualitative aspect of the conclusions. We present our work as a technical note because we aim to provide a quantitative measure for how quickly the dissipation of information in the river network occurs leading to the underlying issues of non-uniqueness in the solution of runoff generation and transport in river networks.

**Note: In our formulas for the parameterization of velocity we use the lower-case letter $q$ for discharge, while Beven uses capital case $Q$ for discharge. In the derivations below the two letters are used interchangeably.**

According to Beven 1979

$$\frac{\partial Q}{\partial t} + C \frac{\partial Q}{\partial x} = C \, q_{lef}(x)$$

where $C$ is celerity (speed of propagation of disturbances)

Our equation derived from continuity and $V = V_o \left(\frac{q}{q_{ref}}\right)^{\lambda_1} \left(\frac{A}{A_{ref}}\right)^{\lambda_2}$ where $A$ is upstream basin area

$$\frac{\partial Q}{\partial t} = \frac{V_o \left(\frac{q}{q_{ref}}\right)^{\lambda_1} \left(\frac{A}{A_{ref}}\right)^{\lambda_2}}{(1 - \lambda_1) \, \ell} \left(q_{lat} - Q + \sum Q_{up}\right)$$

Note that our equation integrates over the reach length ($\sim 500\,m$), but can be rewritten as

$$\frac{dQ}{dt} = \frac{V_o \left(\frac{q}{q_{ref}}\right)^{\lambda_1} \left(\frac{A}{A_{ref}}\right)^{\lambda_2}}{(1 - \lambda_1)} \left(\frac{[q_{lat} + \sum Q_{up}] - Q}{\ell}\right)$$

the term in the parenthesis can be interpreted as $\frac{\partial Q}{\partial x}$

& in our study $\lambda_1 = 0.3$
therefore $\underline{C = 1.43 \, V}$

$$\Rightarrow \quad C = \frac{V}{1 - \lambda_1}$$

**Reviewer # 2 (Dr. Warrick Dawes)**

This article deserves more prominence than being seen as simply technical. It addresses a fundamental issue that arises from increasing computing power and remote data collection toward highly distributed modelling systems with extremely fine spatial and temporal resolution that are sadly, and inevitably, over parameterised. Hydrological modelling results are routinely described with many decimal places and far too many significant digits given the errors inherent in even measuring the target variable, added to errors in model structure and parameter estimation or measurement.

We thank Dr. Dawes for his review of our work. We think that our work contributes to providing practical measures for how quickly information dissipates in the kind of self-similar networks that drain our landscapes. We have tried to connect our work with historical qualitative results that have described how the process of aggregation and attenuation in river networks blurs the role of hillslope scale processes.

I was reminded immediately of an article by Kirchner (2006) from nearly 20-years ago whose title completely embodies the spirit of this article: that we should strive for the correct answer for the correct reason. Advances in AI/ML along with a plethora of physical and empirical models at various levels of complexity show that any answer can be arrived at by many different means without anyone knowing of any of them is doing it correctly.

We have cited this work in the context of the goals of our paper and how it contributes to our understanding of identifying the "right reasons" to describe streamflow generating processes and variability. See line 105

I agree with every word in this manuscript and can only paraphrase: more powerful computers fuelled by higher resolution data are a good servant but a poor master.

We could not agree more with your statement.

Kirchner, J. W. (2006) Getting the right answers for the right reasons: Linking measurements, analyses, and models to advance the science of hydrology. Water Resources Research, 42, W03S04, doi:10.1029/2005WR004362

We have added this reference to our manuscript. See line 105

---

## Author Response (AR2)

**Response to Reviewer Comments and Recommendations for:**

**Technical Note: Testing the Connection Between Hillslope Scale Runoff Fluctuations and Streamflow Hydrographs at the Outlet of Large River Basins**

Ricardo Mantilla[1], Morgan Fonley[2], and Nicolás Velasquez[3]

[1]Department of Civil Engineering, University of Manitoba, Winnipeg, MB, Canada
[2]Alma College, Alma, MI, USA
[3]Iowa Flood Center, University of Iowa, Iowa City, IA, USA

*Correspondence to*: Ricardo Mantilla (ricardo.mantilla@umanitoba.ca)

**MS No.: hess-2023-187**

**Dear Editorial Team,**

In the text below we quote the reviewer's comments verbatim (i.e., including typographical and grammatical mistakes) and we follow them with our responses in blue.

**We have made point by point responses to your comments. We have copied your comments (in italics) followed by our response**

**Comments by Reviewer # 1 (Dr. Keith Beven)**

Just very minor corrections

L69 / L206. Beven and Kirkby should be Beven and Wood, E. F.

L110 celerity

Beven references out of order in reference list.

We have fixed the references list as requested

**Reviewer # 2 (Dr. Warrick Dawes)**

The content of this technical note remains strong after some revision, and I endorse the results and conclusions of it.

• Some of the references got out of order, with the new Bevan citations not grouped with the original Bevan reference.

• The citation for De Boer-Euser et al (2017) has the special character "&endash;" misspelled so it appears as text.

• The citation for Gupta et al (2006) looks odd, as I doubt that the American Cancer Society are the publishers of the book. According to the DOI link provided, the citation is "Gupta, H.V., Beven, K.J. and Wagener, T. (2006). Model Calibration and Uncertainty Estimation. In Encyclopedia of Hydrological Sciences (eds M.G. Anderson and J.J. McDonnell). https://doi.org/10.1002/0470848944.hsa138", with any necessary adjustments for journal requirements.

• The citations of Madsen et al (2020) and Velasquez et al (2021) look incomplete without DOI.

We have corrected the list of references as indicated.